# Semantic Segmentation of Extraocular Muscles on Computed Tomography Images Using Convolutional Neural Networks

**DOI:** 10.3390/diagnostics12071553

**Published:** 2022-06-26

**Authors:** Ramkumar Rajabathar Babu Jai Shanker, Michael H. Zhang, Daniel T. Ginat

**Affiliations:** 1Department of Radiology, University of Chicago, Chicago, IL 60615, USA; rbramkumar@gmail.com (R.R.B.J.S.); michael.zhang@uchospitals.edu (M.H.Z.); 2Department of Radiology, Section of Neuroradiology, University of Chicago, Chicago, IL 60615, USA

**Keywords:** CT, semantic segmentation, extraocular muscles, deep learning, convolutional neural networks, dice coefficient

## Abstract

Computed tomography (CT) imaging of the orbit with measurement of extraocular muscle size can be useful for diagnosing and monitoring conditions that affect extraocular muscles. However, the manual measurement of extraocular muscle size can be time-consuming and tedious. The purpose of this study is to evaluate the effectiveness of deep learning algorithms in segmenting extraocular muscles and measuring muscle sizes from CT images. Consecutive CT scans of orbits from 210 patients between 1 January 2010 and 31 December 2019 were used. Extraocular muscles were manually annotated in the studies, which were then used to train the deep learning algorithms. The proposed U-net algorithm can segment extraocular muscles on coronal slices of 32 test samples with an average dice score of 0.92. The thickness and area measurements from predicted segmentations had a mean absolute error (MAE) of 0.35 mm and 3.87 mm^2^, respectively, with a corresponding mean absolute percentage error (MAPE) of 7 and 9%, respectively. On qualitative analysis of 32 test samples, 30 predicted segmentations from the U-net algorithm were accepted while 2 were rejected. Based on the results from quantitative and qualitative evaluation, this study demonstrates that CNN-based deep learning algorithms are effective at segmenting extraocular muscles and measuring muscles sizes.

## 1. Introduction

Computed tomography (CT) imaging of the orbit with measurement of extraocular muscle size can be useful for diagnosing and monitoring thyroid eye disease and other conditions that may affect the extraocular muscles. To assess the size of the extraocular muscles, parameters such as the muscle diameter, cross-sectional area, and muscle volume have been measured on CT and MRI images [1]. However, the manual measurement of extraocular muscle size is time-consuming and tedious. Automated techniques for segmenting extraocular muscles and estimating muscle sizes can provide a reliable and accurate method for clinical use.

Several techniques to carry out the automated segmentation of extraocular muscles have been developed. Some of the earlier works either relied on operators for manual inputs [2] or required the scan template to be aligned in a rigid manner [3,4]. Xing et al. [5] proposed carrying out segmentation using super pixels, which are groups of pixels with coherent intensities and spatial locations. This approach relies on specific spatial connections and prior knowledge which may not be generalizable. Furthermore, all the above methods [2,3,4,5] were developed and evaluated on magnetic resonance (MR) images. Thus, it is of interest to develop fully automated and generalizable segmentation techniques as an aid to radiologic diagnosis from computed tomography (CT) images.

Machine learning can be used as an aid for detecting abnormalities in imaging, but there is limited medical literature regarding the use of deep learning to achieve clinically applicable segmentation of extraocular muscles in humans. In recent years, deep learning, the subfield of machine learning that uses multilayered neural networks, has shown promising results in many cognitive tasks including the semantic segmentation of medical imaging datasets. The deep learning algorithm proposed by Ronneberger et al. [6] has been widely adopted and improved the effectiveness of convolutional neural networks (CNNs) for semantic segmentation tasks in medical imaging. Milletari et al. [7] proposed the V-net, which is based on a volumetric convolutional neural network. The V-net performs segmentation on three-dimensional image volumes instead of two-dimensional image slices and thereby benefits from utilizing information across slices. Further convolutional neural network architectures have been proposed for the segmentation of other organs and tissues such as lungs, brain regions, and tumors [8,9,10,11].

For extraocular muscle segmentation, Zhu et al. [12] proposed a three-dimensional volumetric convolutional neural network that is based on V-net architecture. This proposed CNN model inputs a volume of thirty-two adjacent slices with cropped region-of-interest, which is localized to the area of either the left or right orbit, of size 256 by 256 pixels. However, this was developed and evaluated only on orbital images acquired without contrast enhancement. Furthermore, depending on the window settings used, the boundaries of extraocular muscles can be subjective and vary between studies. Hanai et al. [13] proposed multiple CNN models where the first CNN segments the globe from the coronal CT image and the second CNN performs the segmentation and trimming of the orbital area.

Thicknesses, cross-sectional areas, and volumes of extraocular muscles can be useful in assessing their size for enlargement and monitoring size differences from progression or response to therapy. These size parameters can vary based on the settings and methods used including window settings and the plane of measurement [14,15]. Since the superior rectus and superior levator palpebrae muscles could not be reliably distinguished from each other, they were measured together as a single muscle group, namely the superior muscle group. To measure the thickness, the horizontal diameters of the lateral and medial rectus, and vertical diameters of the superior group and inferior rectus muscle are used. While the vertical diameters of the superior muscle group and inferior rectus muscle were measured on the coronal plane, the horizontal diameters of the medial and lateral rectus muscles were measured on either the coronal or axial plane. Since the cross-sections of the medial and lateral rectus muscles can be at an angle to the coronal plane, the horizontal diameters as measured on the axial plane may be different from those measured on the coronal plane. On the other hand, cross-sectional areas and volumes are computed directly from the outlined segmentations [1,16]. To compute the cross-sectional areas of the extraocular muscles, the outlined muscle boundaries on the coronal slice and the enclosed pixel sizes are used. Similarly, the muscle volume is computed by adding the previously identified cross-sectional areas and multiplying them with the slice thickness.

Despite the promise deep learning offers, challenges remain in developing clinically applicable algorithms to segment and measure the size of extraocular muscles on CT in an automated and generalizable manner. In particular, the CNN algorithms should be able to carry out the segmentation and muscle size measurements on the overall CT of the orbit including slices and sites that may or may not contain extraocular muscles. Since there can normally be slight asymmetries between the left and right side, radiologists evaluate and record measurements of extraocular muscles separately for left and right orbital areas within their reports and findings. The automated algorithms therefore need to provide segmentations and size results specific to left and right sides, respectively. In the clinical setting, these specific measurements can potentially help radiologists while drafting impressions and findings within their reports.

Here we address these challenges with a deep learning approach that can perform a fully automated segmentation and size measurement of extraocular muscles on CT images of the orbit without any manual inputs from a radiologist. We achieve this by (i) training a convolutional neural network to segment extraocular muscles from orbit CTs, (ii) algorithmically calculating the two-dimensional parameters for muscle size such as thickness and cross-sectional area, and (iii) providing the segmentations and size measurements for left and right side separately. We evaluate the effectiveness of predicted segmentation and measurements by comparing them against their ground truths using both quantitative and qualitative evaluations.

## 2. Materials and Methods

### 2.1. Convolutional Neural Network (CNN)

Convolutional neural networks are a class of neural networks that perform well with data that has a grid-like topology, which in our case is a multidimensional pixel array of CT intensity values [17]. It is composed of multiple sequentially applied convolution operations with each operation expressed as,
(1)s=(x∗w),
where, in CNN terminology, *x* is referred to as the input, *w* as the kernel, and the output *s* as the feature map. For a 2-dimensional input image *I* and 2-dimensional kernel *K* (with dimensions *m* and *n*), the convolution operation resulting in output matrix *Z*, implemented mathematically using a cross-correlation operation, can be formulated as,
(2)$$Z\left(i,j\right)=\left(K*I\right)\left(i,j\right)=\underset{m}{\sum }\underset{n}{\sum }I\left(i+m,\;j+n\right)K\left(m,n\right)$$
where *i* and *j* represent the element in the *i*th row and *j*th column of the matrix.

A typical layer in a convolutional network as shown in Figure 1 comprises three operations: the convolution operation, the activation function followed by the pooling function. The convolution operation applies many kernels (*K_i_*) so that many different feature maps are extracted at each layer. The activation function inputs the feature map (*Z*) from the convolution operation and outputs the non-linear activation (*A*) thereby introducing non-linearity in the layer. Most recent CNNs use the rectified linear unit (ReLU) as an activation function, which is an element-wise operation on the feature map *Z*, expressed as *a = max(0,s)*. This is followed by a pooling function which replaces the output of the network at a certain location with a summary statistic of the nearby outputs, which helps make the pooling output (*P*) approximately invariant to minor translations or scale of the input.

Convolutional layers progressively extract higher dimensional image representations (*P^l^*—output *P* at layer *l*). With enough layers and training, a deep convolutional network can yield robust features that help perform the cognitive task. The aim of training is to arrive at the optimized set of values, referred to as parameters, within the kernels *K^l^* (kernel *K* at layer *l*). These parameters can successfully transform the original CT slice/volume (*I*), with values in Hounsfield units (HUs) into the regions-of-interest, i.e., segmentation masks for extraocular muscles. Arriving at this set of optimized network parameters is an optimization task that is carried out using gradient-based algorithms such as gradient descent, which iteratively updates the network parameters to arrive at the final network weights.

The semantic segmentation of images involves assigning a class label to each pixel in the image [18]. In the context of medical images, it can be used to segment anatomical tissues which can later be analyzed for diagnosis purposes. Prior to deep learning, semantic segmentation was carried out using pixel-wise classifiers such as random forests [19], where the prediction for a specific pixel was made using the pixel intensities around that pixel. Several convolutional neural network architectures have been shown to be useful in medical image segmentation in recent years. While there have been many individual architectures proposed, the existing CNN based medical image segmentation architectures can be classified into three categories: fully convolutional neural networks, U-net, and generative adversarial networks [20].

Fully convolutional networks (FCNs), proposed by Long et al. [21], was one of the first deep learning works for semantic segmentation that used only convolutional layers. This CNN architecture takes an image of any size and applies a series of successive convolutional operations and produces the output segmentation map with the same size as the input image. While a typical convolution operation would result in an output feature map that is of lower size than input image, the final two layers use a deconvolution layer that up-samples the feature map from the previous layer and outputs the resulting image with same size as the original image. This architecture also uses skip connections where the output from initial layers of the model is combined with the inputs to the final prediction layer to provide higher-level semantic information, which results in better predictions that respect global structure. However, the results from the up-sampling layers in FCN were still relatively fuzzy and insensitive to the details. These shortcomings were addressed in the subsequent CNN architectures such as the DeepLab v1, DeepLab v2, DeepLab v3, and DeepLab v3+ [22,23,24,25] which resulted in better segmentation boundaries and at multiple scales and made use of conditional random fields (CRF) [26]. SegNet [27] used the encoder-decoder architecture with the up-sampling operation performed by a trainable convolution layer. FCN networks have been used to segment multiple organs and tissues such as brain tumors [28,29,30], eye [31,32], chest [33], liver [34], and left and right ventricles of the heart [35].

The U-net architecture, proposed by Ronneberger et al. [6], is based on the encoder-decoder architecture where the encoder module (contracting path) captures context, and the expanding decoder module (expanding path) enables precise localization. The 3D U-net, proposed by Çiçek et al. [36], realizes 3D image segmentation by inputting a continuous sequence of 2D images. The V-net [7] can perform segmentation on 3D volumes by using 3D convolution kernels in place of 2D convolution kernels. Further works improved on the U-net by adding an attention mechanism that helped the network localize better [37]. U-net and its variants have been used to segment multiple organs and tissues including retinal vessels [38], chest [39], and heart [40].

Generative adversarial networks (GANs) are a class of neural networks in which two networks, the generator module and discriminator module, compete against each other. While the generator network uses random noise to generate an image, the discriminator network judges whether the image is “real” or not. As iterations progress, the generator network gets better at generating images that look more real and the discriminator network becomes better at judging the generated images. In the work proposed by Luc et al. [41], the generator network generated segmentation maps, and the discriminator network judged whether the segmentation maps were coming from the ground truth or the generator. GANs have been shown to successfully segment the brain [42], retinal vessels [43], and spines [44] from medical images.

Due to its excellent performance, the U-net and its variants have been widely used in various fields of computer vision. The U-net was chosen for this implementation because of its ability to capture global context and precise localization. As shown in Figure 2, we use the U-net architecture with convolutional layers as displayed in the legend (bottom-right). The network is comprised mainly of down-sampling layers, i.e., convolutional layers that reduce the feature map size, and up-sampling layers, i.e., convolutional layers that increase the feature map size, and skip connections between them to provide a direct flow of feature maps from an early layer in the network to a later one. Skip connections are realized by either concatenating the feature maps of the early layer to those of the later one or by applying an element-wise summation.

### 2.2. Dataset

For this retrospective study, we analyzed coronal CT images of the orbit acquired from 215 patients between 1 January 2010 and 31 December 2019. After excluding patients with facial trauma and/or image artifacts, the final 210 patients were randomly split into training and test sets with 178 and 32 patients, respectively. The training set was used to develop the model and the test set was used to evaluate the performance of the model on scans previously unseen to the model. The model’s performance on a test set is important since it gives an estimate of its ability to generalize predictions to unseen scans.

Each scan was annotated using 3D slicer (version 4.11) to create masks (ground truths) of extraocular muscles from the DICOM files [45]. A multiclass mask was created with muscles classes (L-medial rectus, L-lateral rectus, L-superior group (including L-superior rectus and L-superior levator palpebrae), L-inferior rectus, R-medial rectus, R-lateral rectus, R-superior group (including R-superior rectus and R-superior levator palpebrae), R-inferior rectus) and background class.

The ground truths for extraocular muscle segmentations were analyzed by a board-certified radiologist with a certificate of added qualification in neuroradiology. These ground truths were used to train the CNNs in a supervised manner. Figure 3 shows an example of (L-R) coronal plane, axial plane, and sagittal plane with the original DICOM image (left) and EOM masks highlighted.

The comparison (Table 1) between the training and test set on baseline patient characteristics (age, sex, and ground truth EOM thickness measurement) shows that there are no significant differences between the training set versus the test set.

### 2.3. Image Acquisition

The orbit scans were performed using 65 mL of Omnipaque 350 (injection rate of 1.2 mL/s with a delay of 55 s). Image acquisition was performed with field of view of 200 mm, collimation of 64 by 0.625 mm, source slice thickness of 0.9 mm with an increment of 0.45 mm, 120 kV, 200 mAs, and 3 mm soft-tissue reconstructions in axial, coronal, and sagittal planes.

### 2.4. Data Preprocessing

Since different studies may have varying different pixel spacing values, they are first isometrically resampled to pixels of size 1 × 1 mm with the aid of the PixelSpacing DICOM attribute and spline interpolation of order three. To facilitate a standard input size for the network, we used a patch-based input as used in the original implementation of U-net. For example, using this method, multiple patches of size (64, 64 pixels) are drawn from random areas of an original CT slice of size (512, 512 pixels). This method also acts as a data augmentation process and can alleviate the challenges of using a small dataset. Since the patches can be drawn from any area of the overall image, the network learns to be translation invariant and can effectively segment extraocular muscles in any localized area of an input slice. Further data augmentation methods, which improve the robustness of the model such as rotation, size scaling and noise addition are applied. Flipping images horizontally from left right was not performed in this work. This was to enable the network to learn to differentiate extraocular muscles on the left orbit from those on the right. Noise addition to CT images that can simulate low-dose acquisition settings require access to the raw scanning data [46,47]. Since the raw sinogram data from the scanners was not available at the time of training, Gaussian noise (of mean = 0 and standard deviation = 10 Hounsfield Units) was added to the CT slice intensities. The scans were then windowed to highlight extraocular muscles using level and width settings of 50 and 250 Hounsfield Units, respectively [14]. The scan inputs were then provided as a stack of 2-dimensional patches, which were then fed into the network. Half of the patches in this stack were drawn from the orbit area, and the other half was drawn from non-orbital areas.

### 2.5. Architecture

We employed the 2-dimensional implementation of U-net as our CNN architecture. Since the coronal plane is the only plane in which all rectus muscles can be visualized on a 2-dimensional slice, we trained a 2-dimensional U-net network that could predict all four rectus muscles (left and right) using only coronal slices.

### 2.6. Loss Functions

Loss functions, which measure the dissimilarity between actual and predicted segmentations, are important because they guide the network to learn meaningful predictions. They also govern how the network should learn from mistakes (false positives, false negatives, segmentation boundaries vs. volume, hard vs. easy scenarios). Loss functions can be formulated to measure the mismatch in distribution, region, boundary, or a combination of these [48]. Distribution-based loss functions such as weighted cross entropy (WCE) train the network to minimize dissimilarity between the predicted and ground truth distributions. Region-based loss functions such as the Dice similarity coefficient loss [49], Jaccard (Intersection over Union) loss [50], and Focal Tversky loss [51] aim to minimize the mismatch or maximize the overlap regions between the predicted and ground truth segmentations. Boundary-based losses such as the boundary loss [52], surface Dice similarity coefficient [10] aim to minimize the difference between the contours of predicted and ground truth segmentations.

Weighted cross entropy (WCE) loss is defined as the measure of difference between two distributions and is given mathematically defined as
(3)$$WCE\;loss=-\frac{1}{N}\;\sum_{c=1}^{C}\sum_{i=1}^{N}w_{c}g_{i}^{c}\log s_{i}^{c},$$
where *g_i_^c^* is the ground truth value, *s_i_^c^* is the corresponding predicted segmentation probability and *w_c_* is the weight for each class *c*, and *N* is the total number of pixels.

Dice similarity coefficient (DSC) loss, also known as the overlap index, is used to compare the similarity between predicted and ground truth segmentations. For binary labels of ground truth *g_ic_* ∈ {0,1} and predicted probability of class label *p_ic_* ∈ [0,1] with total number of pixels N, the DSC loss is expressed as
(4)$$DSC\;loss=1-\frac{\sum_{i=1}^{N}p_{ic}g_{ic}+\varepsilon }{\sum_{i=1}^{N}p_{ic}+g_{ic}+\varepsilon },$$
where *ε* is a smoothing parameter with a value close to zero, which provides numerical stability to prevent division by zero. Lower values of *DSC* indicate better overlap between the predicted and ground truth segmentations.

Jaccard (Intersection over Union) loss measures the extent of overlap and penalizes regions that do not overlap with the ground truth. For binary labels of ground truth *g_ic_* ∈ {0,1} and predicted probability of class label *p_ic_* ∈ [0,1] with total number of pixels *N*, the *IOU loss* is expressed as
(5)$$IOU\;loss=1-\frac{\sum_{c=1}^{C}\sum_{i=1}^{N}g_{i}^{c}p_{i}^{c}}{\sum_{c=1}^{C}\sum_{i=1}^{N}g_{i}^{c}+p_{i}^{c}-g_{i}^{c}p_{i}^{c}},$$

Focal Tversky loss is an extension of the dice loss and addresses some of the issues with Dice loss where the regions-of-interest are small. The Tversky index, in which generalization of Dice loss that allows for flexibility in balancing false-positives and false-negatives., is combined with the γ parameter, which controls for easy background and hard ROI training examples. The Focal Tversky loss is given by,
(6)FTLc=∑c(1−TIc)1γ,
where *TI_c_* is the Tversky Index, and parameter γ varies in the range [1,3].

Boundary loss measures the distances between two boundaries. The integral framework, which is differentiable and can be used as a loss function, is formulated as
(7)$$L_{BD}=\sum_{\Omega }\phi_{G}\left(p\right)s_{\theta }\left(p\right),$$
where *ϕ_G_* is the level set representation of the ground truth boundary and $s_{\theta }\left(p\right)$ is the SoftMax probability outputs from the trained network.

Compound loss functions use a combination of the above losses that are tailored for a specific application. For example, if the objective of the segmentation is to arrive at only volume or area measurements, region-based loss functions would be well suited to train the CNN. However, if the clinical objective is to identify organ contours for radiation therapy or a thickness measurement which depends on the contours identified, distance-based loss functions would be well suited. Boundary-based losses would need to be used jointly with a region-based loss to deliver improved segmentation results. Compound losses have also been shown to be more robust loss functions [48].

It is worth noting that, in our application with extraocular muscles, we measure not only the area and volume of the predicted segmentation but also its thickness, which is measured from muscle boundary contours. For this reason, we train the U-net using not only individual loss functions but also compound loss functions and pick the best performing model.

### 2.7. Training & Experiment Design

To train our model, we used an Adam optimizer [53]. Adam is an optimization algorithm that is used to arrive at the final network weights from a set of randomly initialized weights by updating the network weights in an iterative manner. The networks were allowed to train for a maximum of one hundred epochs on a Nvidia Tesla K80 GPU (maximum 8 h).

Hyperparameters are model parameters that are set to control the learning process. While some hyperparameters were chosen based on default values, other values were chosen after preliminary performance evaluation.

Learning rate: Learning rates control how much the model weights are updated at the end of each iteration. A large learning rate helps the algorithm learn fast, but it may also make the training process unstable. On the other hand, a small learning rate will require many updates before reaching the optimal solution and may make the training process take too long to converge. Since Adam is an adaptive optimization algorithm, the learning rates are computed individually for different parameters from estimates of first (mean) and second (uncentered variance) moments of the gradients. After a preliminary evaluation of the default settings, we set the value for learning rate of 0.0001, exponential decay rate for the 1st moment (mean) estimates as 0.9, and exponential decay rate for the 2nd moment (uncentered variance) estimates as 0.999.

Image patch size: Patch size is the size of each image input to the CNN during the training process. While larger patches contain more information, they also take more memory resources during training. The U-net architecture applies a series of four pooling operations, each of which reduce the image size by a factor of two. Therefore, the minimum input image size would be 16. However, image patches of size 16 × 16 pixels would not contain enough spatial and contextual information for the network to train on. On the other hand, the maximum patch size is constrained by the maximum orbital scan size in the coronal plane. In this work, a patch size of 128 × 128 pixels was used to facilitate the patches generated to have at least one orbit fully with adjoining areas.

Batch size: Batch size is the number of patches that are input to the CNN at each step of the training iteration. Using smaller batch sizes makes the network more resilient to noise but also increases the training time significantly. To make efficient use of the GPU memory, we used a batch size of 20 patches drawn from 4 training images for each step within the training iteration.

Loss functions: In this work, we evaluated the performance for individual loss functions and compound loss functions below: (i) weighted cross entropy, (ii) Dice similarity coefficient loss, (iii) weighted cross entropy + Dice similarity coefficient loss, (iv) Focal Tversky loss, (v) Dice similarity coefficient loss + boundary loss.

Dropout: Dropout [54] is a technique that approximates training many networks with different architectures in parallel. This is implemented during training by randomly setting a portion of the network to zero, thus having the effect of making the architecture be treated as a layer with different numbers of nodes. This also helps prevent overfitting by making the training process noisier and breaking-up co-adaptation situations where hidden units may change in a way that they fix up the mistakes of other units. Dropout has a tunable hyperparameter *p*, which is the probability of retaining a unit in the network. This hyperparameter controls the intensity of dropout, where higher values of *p* (*p* ≈ 1) correspond to lower dropout and lower values (*p* ≈ 0) correspond to more dropout. In this work, we set the dropout hyperparameter *p* as 0.8.

Weight initialization: Initializers are used to define the way to set the initial random weights of CNN layers. Initialization can have a significant impact on the training process, convergence, and final performance. While a network initialized with high initial weights may lead to exploding gradients, a network initialized with too low initial weights may lead to vanishing gradients. In this work, we used Glorot normal initialization, also known as Xavier initialization [55], where the weights are initialized by drawing samples from a truncated normal distribution centered on zero and standard deviation, which is computed dynamically as *sqrt(2/(fan_in + fan_out)),* where *fan_in* is the number of input units in the weight tensor and *fan_out* is the number of output units in the weight tensor.

Cross-validation is a resampling method that uses different sections of the data to train and validate the model. As illustrated in Figure 4, we performed k-fold cross validation, where the original training sample of 178 studies is partitioned into *k* equal sized subsamples. Of the *k* subsamples, a single subsample is chosen as validation set for evaluating the model performance, and the remaining *k−1* subsamples are used as training set. This process is repeated *k* times, with each subsample used exactly once as the validation set. In our experiments, we set *k* as 10. The validation results were then aggregated to produce a single estimation. Using this estimate, we could compare the performance across parameters, and select the best performing model. After this step, the best parameters were then used to train the model on the training data and then the final evaluation was carried out on test data.

### 2.8. Muscle Size Measurement

For a given coronal slice *i*, the thickness of the extraocular muscles is measured by fitting a rotated rectangle of minimum area that completely encloses the segmented muscle contour. Since the cross-sections of extraocular muscles (in the mid-orbital region) on the coronal plane are nearly ellipsoids, the length and width of the bounding rectangle are the long axis diameter and short axis diameter (or thickness), respectively, of the muscle cross-section.

As illustrated in Figure 5, the thickness (*t*) for each muscle *m* on coronal slice *i*, is calculated as*t_m_^i^ = width (R_m_^i^),*(8)
where *R_m_^i^* is the bounding rectangle with minimum area of muscle *m* on slice *i*. The maximum thickness across all coronal slices is taken as the overall thickness of the muscle.

Similarly, the cross-sectional area (*A*) of muscle *m* on coronal slice *i* is given by*A_m_^i^ = N_m_^i^*,(9)
where *N_m_^i^* is the number of pixels within the segmentation of muscle *m* on coronal slice *i*. The maximum cross-sectional area across all coronal slices is taken as the cross-sectional area of the entire muscle.

### 2.9. Evaluation

To perform quantitative evaluation, the Dice coefficient and intersection-over-union (IOU) metrics were used. The extraocular muscles originate from the common tendinous ring located at the apex of the orbit and insert onto the sides of the eyeball. The cross-sections of the extraocular muscles become small as they crowd together towards their ligamentous origin. Therefore, it is especially challenging for automated algorithms to reliably segment the rectus muscles from each other near the apex of the orbit. Similarly, at the anterior aspect of the orbit, it is hard to distinguish extraocular muscles from the tendons inserting into the globe. For this reason, we split the coronal slices of the extraocular muscles into three sections—(i) near insertion, (ii) central part, and (iii) near tendinous origin. As shown in Figure 6, this was achieved by splitting the coronal slices in each study into three equal parts to form the three regions. We present an evaluation of the model performance specific to each region.

We also compared the thickness and cross-sectional area measurements of extraocular muscles from ground truth segmentation to those from predicted segmentations using the two metrics—mean absolute error (MAE) and mean absolute percentage error (MAPE), given by
(10)$$\mathrm{MAE}=\frac{1}{n}\overset{n}{\underset{i=1}{\sum }}\left|x_{i}^{pr}-x_{i}^{gt}\;\right|$$
(11)$$\mathrm{MAPE}\;=\frac{1}{n}\overset{n}{\underset{i=1}{\sum }}\left|\frac{x_{i}^{pr}-x_{i}^{gt}\;}{x_{i}^{gt}}\right|$$
where *x_i_^pr^* is the predicted value, *x_i_^gt^* is the ground truth value of the *i*th sample, and n is the number of samples. The MAE and MAPE together can be used to evaluate how closely the predicted measurements from the model align with the ground truth measurements.

To perform qualitative evaluation, the predicted segmentations from the models were analyzed visually by the radiologist for similarity of contours and other subjective assessments. An accept/reject decision was provided with rejected segmentations accompanied by a reason.

## 3. Results

### 3.1. Quantitative Evaluation

To evaluate the performance on the validation and test data, the same data preprocessing steps were applied during model training, i.e., isometric resampling, drawing image patches, and CT windowing, were applied to the test dataset as well. The final predicted slice was then reconstructed from the individual patch predictions using sliding window predictions and majority voting for each pixel to facilitate thickness and area measurements.

#### 3.1.1. Model Performance

The results from 10-fold cross validation training process are summarized in Table 2. The model trained using WCE+Dice compound loss function had the best overall performance on cross-validation data with a mean Dice score of 0.92 for all eight extraocular muscle classes with a standard deviation of 0.03. This was followed by the U-net model trained using Dice+Boundary compound loss function with a mean dice score of 0.91 and a standard deviation of 0.04. While the Dice+Boundary compound loss function had a better Dice score for L-medial rectus and L-lateral rectus, the WCE+Dice compound loss function consistently outperformed the other loss functions in the remaining classes. For this reason, we selected the model trained using WCE+Dice compound loss as the final model.

The performance of the selected U-net model was evaluated using test data and the results are summarized in Table 3. The selected model that was trained using WCE+Dice loss function could segment the extraocular muscles with a mean Dice score of 0.92 and a standard deviation of 0.02, which corresponds to an IOU score of 0.87 and a standard deviation of 0.03.

Model performance was also evaluated on different regions of the extraocular muscles near their origin, central part and near their insertion and the results summarized in Table 4. We observed that the model performed better on the central region than the regions near the origin and insertion but the difference in performance was not statistically significant (*p*-value = 0.1763).

#### 3.1.2. Comparison of Muscle Size Measurements

The results from thickness and area measurements are summarized in Table 5. The thicknesses and areas measured from the predicted segmentations, when compared with the ground truth segmentations, had a mean absolute error of 0.35 mm and 3.87 mm^2^, respectively. The corresponding mean absolute percentage errors in thickness and area were 7 and 9%, respectively.

#### 3.1.3. Model Performance on Noisy Images

To assess its robustness, the trained model was evaluated with inputs with varying degrees of noise. During the training stage, the model was trained using images added with a Gaussian noise with zero mean and a standard deviation of 10 HUs. During the testing stage, the input images were added with a Gaussian noise with standard deviations of 5 and 10 HUs, respectively. The performance of the trained U-net on noisy images is shown in Table 6. There was no significant reduction in performance (*p*-value = 0.8913), evaluated using the Dice score, of the model on noisy images with Gaussian noise up to (μ = 0, σ = 10). This can be explained by the fact that the U-net was trained on similar noisy images during the training stage. Figure 7, shows a sample coronal patch from the test data with and without noise, and its corresponding predictions by the trained model.

#### 3.1.4. Performance Comparison with Traditional Segmentation Methods

Prior works that used traditional segmentation methods to segment extraocular muscles were built and evaluated on MR images [2,3,4,5]. Hanai et al. [13] proposed a deep-learning method to detect enlarged muscles and therefore only provided classification accuracy of their model in detecting enlarged extraocular muscles and not segmentation evaluation results. The U-net based CNN model proposed by Zhu et al. [12] was built and evaluated on CT images. However, the imaging studies and ground truths that were used varied between the studies. Since there are no established benchmarks for extraocular muscle segmentation, we show the results from our model alongside previous works. In our proposed model, the superior rectus and superior levator palpebrae muscles were measured together as a single muscle group, namely, the superior muscle group. We also consider extraocular muscles from left and right as different classes. The analysis of the evaluation metric (intersection over union) is summarized in Table 7.

### 3.2. Qualitative Evaluation

Among the thirty-two test samples, thirty predicted segmentations from the U-net algorithm were accepted while two were rejected. Example coronal slices from the two rejected segmentations are presented in Figure 8. The predicted segmentation on Study ID 2G04345 was rejected because the L-lateral rectus also included other orbital structures. The predicted segmentation on Study ID 2G04323 was rejected because the L-lateral rectus included areas of bone.

## 4. Discussion

We have developed and evaluated a 2D CNN-based deep learning algorithm that can perform the automated segmentation of extraocular muscles and provide measurements of two-dimensional parameters for muscle size, such as thickness and cross-sectional area. The proposed algorithm provides a method to carry out automated segmentation in a computationally efficient way using only images in the coronal plane of a CT scan.

To improve the segmentation accuracy further, postprocessing steps such as thresholding, erosion and dilation can be applied. The segmentations can therefore be further refined to exclude bone and other orbital structures that are not extraocular muscles. Furthermore, it is worth noting that the thickness errors (in mm) between predicted and ground truth measurements were in the same range as the pixel sizes (0.3–0.4 mm) of the CT images. This could be due to the data preprocessing (isometric resampling) step where we downsampled all CT images to a constant pixel size of 1 mm × 1 mm. Since the CNN algorithm reads and makes the prediction on resampled images, a single-pixel misclassification along the short-axis of extraocular muscles can result in a MAPE of up to 10% during reconstruction back to original pixel size. The downsampling step could potentially result in the loss of granular information and therefore drive the errors in thickness and area measurements. A training methodology that uses the original pixel intensities without isometric resampling and various postprocessing techniques will be explored as part of future work.

Neural networks perform best and generalize successfully when input data at the time of inference has a similar data distribution to that of the training data used. The proposed model was developed using CT images of the orbit and of extraocular muscles (EOM) with and without enlargement for male and female cases. Thus, it may not perform well with scans of other modalities, such as MRI. Furthermore, the model needs to be trained using other types of scans that might not include EOM for it to learn the other organs that do not constitute EOM. Single institution training and testing data were used for this study. Generalizability to other institutions and patient populations should be evaluated in the future.

In this work, we evaluate one type of neural network architecture, i.e., the encoder-decoder architecture to carry out semantic segmentation. As next steps, we could carry out further evaluations using other neural network architectures that have shown promise to work well for semantic segmentation, such as residual networks (ResNets) and region-proposal networks.

## 5. Conclusions

Based on the results from the quantitative and qualitative evaluations, this study demonstrates that CNN-based deep learning algorithms are effective at segmenting extraocular muscles and measuring muscle sizes on CT images without any manual inputs from a radiologist.

## Figures and Tables

**Figure 1 diagnostics-12-01553-f001:**
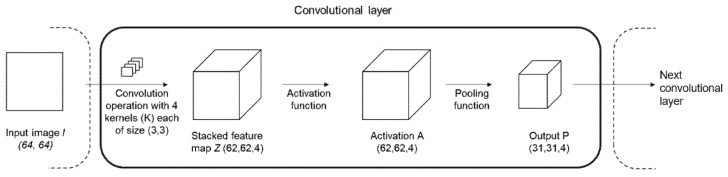
Typical convolutional layer comprising of convolution, activation, and pooling operations.

**Figure 2 diagnostics-12-01553-f002:**
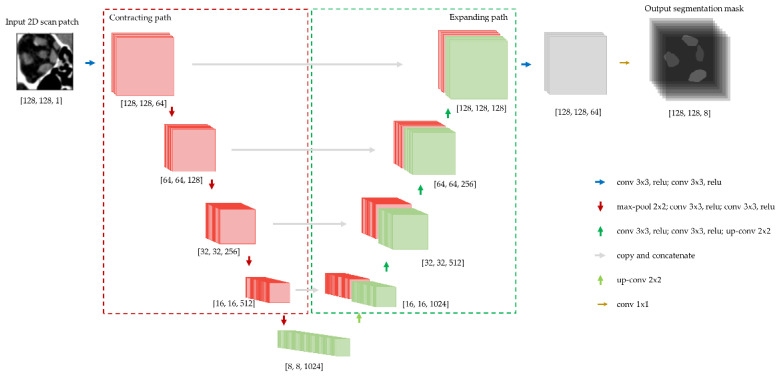
Schematic of U-net network architecture [7] with image/feature map size on bottom of the box (adapted to input patch size of 128 × 128) and convolutional layers depicted using arrows.

**Figure 3 diagnostics-12-01553-f003:**
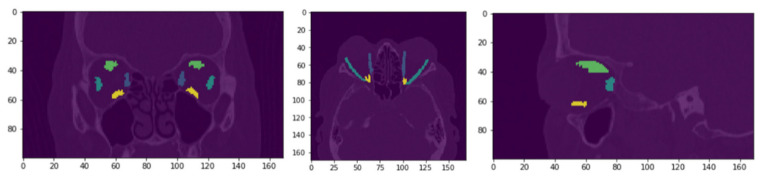
Label-wise annotation of EOMs with EOM masks highlighted on (L-R) coronal, axial, and sagittal CT images.

**Figure 4 diagnostics-12-01553-f004:**
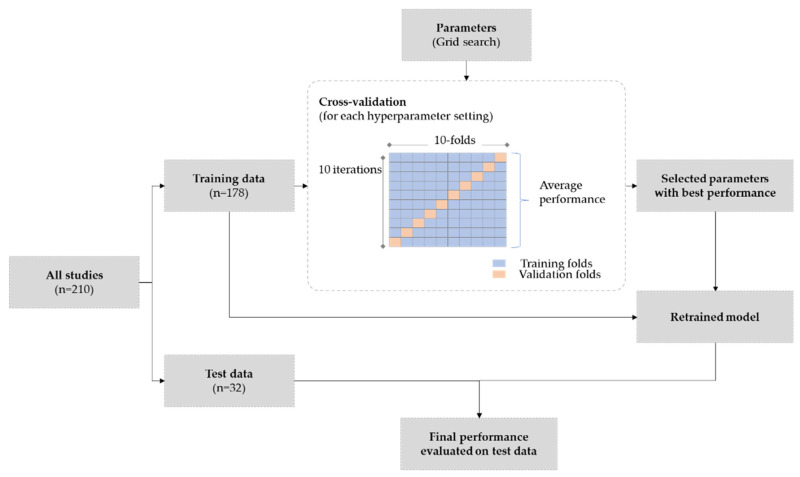
Experimental setup for model training, cross-validation, and final evaluation.

**Figure 5 diagnostics-12-01553-f005:**
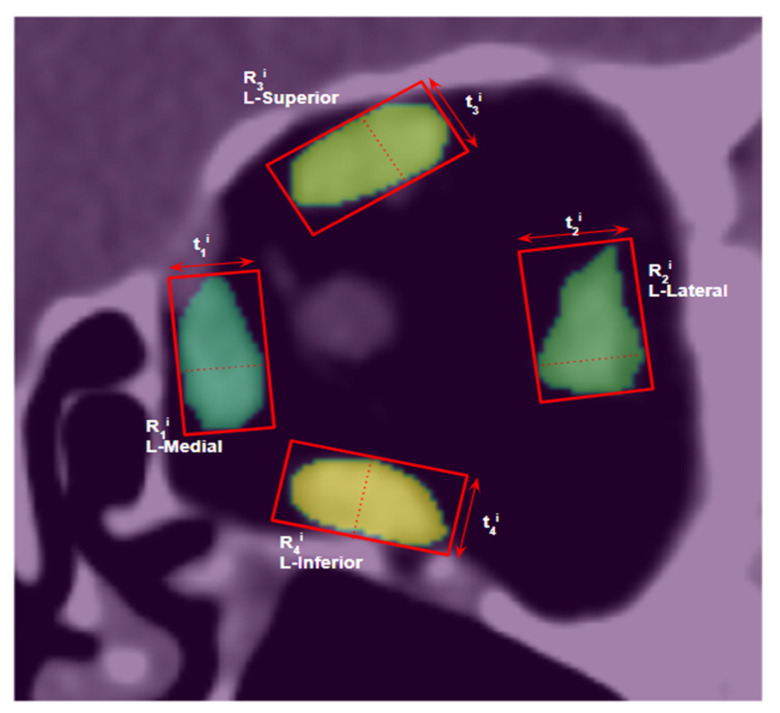
Illustration of thickness measurements (*t*_1_*^i^*, *t*_2_*^i^*, *t*_3_*^i^*, *t*_4_*^i^*) on a coronal slice *i*, using rotated rectangles of the minimum area (*R*_1_*^i^*, *R*_2_*^i^*, *R*_3_*^i^*, *R*_4_*^i^*) enclosing the segmented muscle contours (L-medial, L-lateral, L-superior, L-inferior), respectively.

**Figure 6 diagnostics-12-01553-f006:**
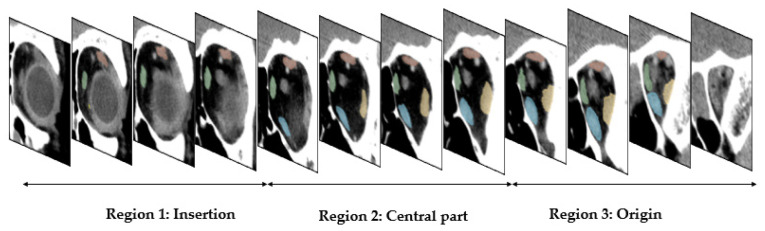
Illustration that shows the coronal slices containing extraocular muscles being split equally into three regions that include the muscle insertion, central part, and origin, respectively.

**Figure 7 diagnostics-12-01553-f007:**
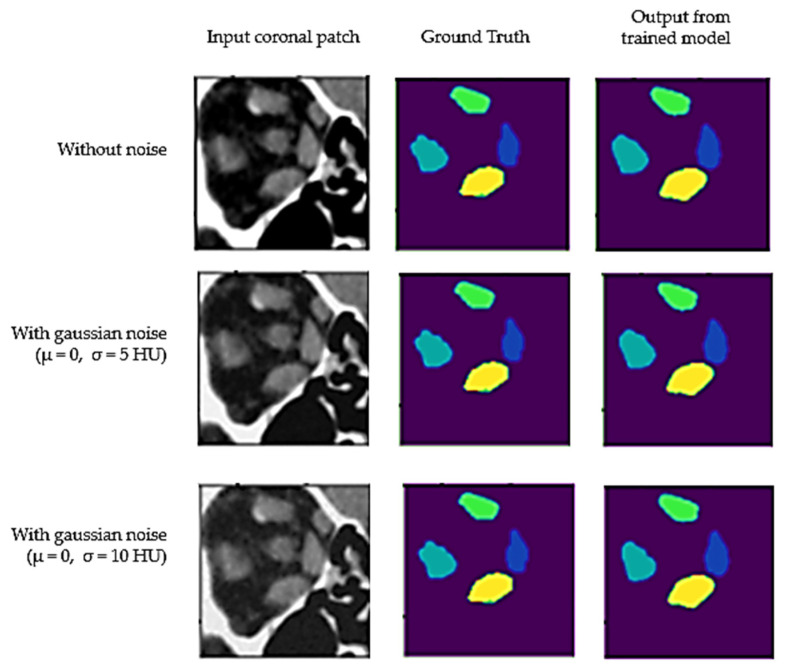
Output from the trained U-net on sample coronal patch where input image has been added with different levels of Gaussian noise (with mean μ and standard deviation σ).

**Figure 8 diagnostics-12-01553-f008:**
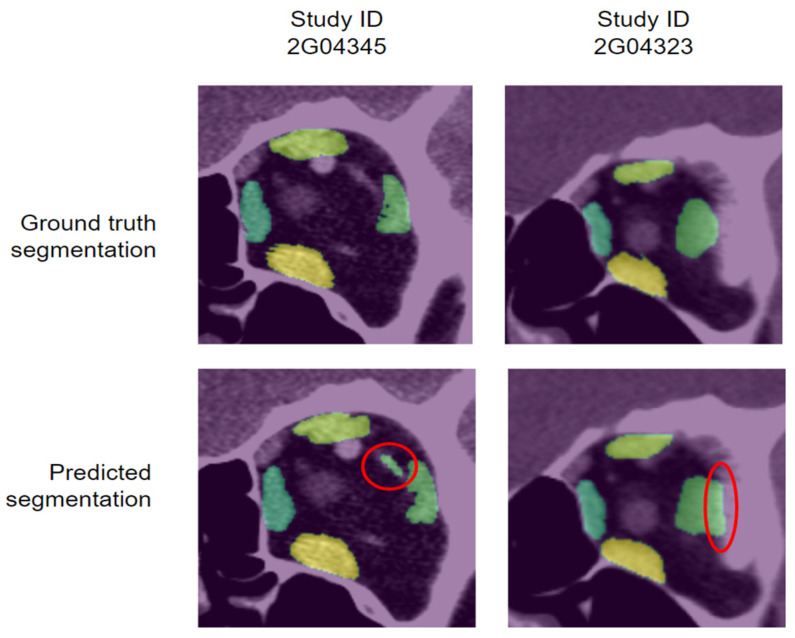
Coronal slice examples from the two test samples whose predicted segmentations were rejected after qualitative evaluation. Red circles outline the areas erroneously predicted by the U-net algorithm.

**Table 1 diagnostics-12-01553-t001:** Baseline patient characteristics of the training and test groups with mean (and standard deviation) thickness and area in mm and mm^2^ respectively.

	Train	Test	*p*-Value
N	178	32	
Sex = M (%)	53 (30%)	9 (28%)	1
Age	46.67 (17.49)	50.97 (19.7)	0.21
Thickness—L-Medial Rectus	4.87 (0.84)	4.85 (0.7)	0.9
Thickness—L-Lateral Rectus	5.5 (1.13)	5.38 (1.18)	0.58
Thickness—L-Superior group	4.79 (0.93)	4.9 (0.77)	0.53
Thickness—L-Inferior Rectus	5.23 (1.07)	5.19 (1.02)	0.84
Thickness—R-Medial Rectus	4.74 (0.66)	4.66 (0.85)	0.55
Thickness—R-Lateral Rectus	5.62 (1.41)	5.87 (1.33)	0.35
Thickness—R-Superior group	4.85 (1.04)	4.99 (0.91)	0.48
Thickness—R-Inferior Rectus	5.13 (1.08)	4.97 (0.98)	0.44
Area—L-Medial Rectus	38.93 (8.54)	39.16 (6.55)	0.89
Area—L-Lateral Rectus	46.03 (10.03)	46.17 (12.05)	0.89
Area—L-Superior group	38.29 (9.66)	40.28 (8.04)	0.27
Area—L-Inferior Rectus	41.57 (11.78)	41.17 (9.35)	0.86
Area—R-Medial Rectus	38.14 (6.88)	38.56 (7.01)	0.75
Area—R-Lateral Rectus	47.2 (14.29)	49.81 (12.33)	0.33
Area—R-Superior group	40.1 (13.79)	41.39 (9.62)	0.61
Area—R-Inferior Rectus	42.38 (14.24)	41.14 (10.8)	0.64

**Table 2 diagnostics-12-01553-t002:** Results from training and evaluation using 10-fold cross-validation. Values indicate mean ± standard deviation of Dice score and IOU score from 10 cross-validation iterations. Values in bold represent the loss function setting that provides the best performance for a specific muscle class.

Evaluation Metric	Muscle	Loss Function				
		WCE	Dice	WCE + Dice	FTL	Dice + Boundary
Dice similarity coefficient (DSC) score	L-medial rectus	0.90 ± 0.01	0.91 ± 0.03	0.93 ± 0.02	0.90 ± 0.05	**0.94 ± 0.01**
	L-lateral rectus	0.90 ± 0.00	0.91 ± 0.04	0.91 ± 0.03	0.90 ± 0.05	**0.93 ± 0.01**
	L-superior group	0.84 ± 0.03	0.90 ± 0.03	**0.91 ± 0.02**	0.87 ± 0.06	0.87 ± 0.05
	L-inferior rectus	0.90 ± 0.02	0.92 ± 0.03	**0.94 ± 0.02**	0.90 ± 0.03	0.93 ± 0.02
	R-Medial rectus	0.90 ± 0.00	0.93 ± 0.02	**0.94 ± 0.01**	0.91 ± 0.02	0.93 ± 0.01
	R-lateral rectus	0.88 ± 0.01	0.91 ± 0.04	**0.91 ± 0.04**	0.88 ± 0.06	0.90 ± 0.05
	R-superior group	0.85 ± 0.01	0.89 ± 0.02	**0.91 ± 0.02**	0.87 ± 0.03	0.88 ± 0.03
	R-inferior rectus	0.91 ± 0.01	0.90 ± 0.04	**0.92 ± 0.02**	0.90 ± 0.05	0.92 ± 0.03
	All	0.89 ± 0.03	0.91 ± 0.03	**0.92 ± 0.03**	0.89 ± 0.05	0.91 ± 0.04
Jaccard (IOU) score	L-medial rectus	0.81 ± 0.02	0.86 ± 0.04	0.88 ± 0.03	0.83 ± 0.07	**0.89 ± 0.02**
	L-lateral rectus	0.83 ± 0.00	0.85 ± 0.05	0.86 ± 0.04	0.84 ± 0.06	**0.87 ± 0.01**
	L-superior group	0.73 ± 0.04	0.82 ± 0.04	**0.85 ± 0.03**	0.79 ± 0.07	0.80 ± 0.05
	L-inferior rectus	0.82 ± 0.03	0.86 ± 0.04	**0.88 ± 0.03**	0.83 ± 0.04	0.87 ± 0.04
	R-medial rectus	0.82 ± 0.00	0.87 ± 0.03	**0.89 ± 0.01**	0.84 ± 0.03	0.88 ± 0.02
	R-lateral rectus	0.79 ± 0.02	0.85 ± 0.05	**0.85 ± 0.05**	0.81 ± 0.07	0.84 ± 0.06
	R-superior group	0.75 ± 0.01	0.82 ± 0.02	**0.84 ± 0.03**	0.79 ± 0.03	0.80 ± 0.03
	R-inferior rectus	0.83 ± 0.02	0.84 ± 0.04	**0.87 ± 0.03**	0.83 ± 0.07	0.87 ± 0.04
	All	0.80 ± 0.04	0.85 ± 0.04	**0.87 ± 0.04**	0.82 ± 0.06	0.85 ± 0.05

**Table 3 diagnostics-12-01553-t003:** Performance of selected U-net model (trained using WCE+Dice loss) on test data. Values indicate mean ± standard deviation of Dice score and IOU score across 32 test scans.

Muscle	DSC Score	IOU Score
L-medial rectus	0.94 ± 0.07	0.90 ± 0.09
L-lateral rectus	0.93 ± 0.09	0.88 ± 0.10
L-superior group	0.90 ± 0.12	0.83 ± 0.13
L-inferior rectus	0.94 ± 0.08	0.90 ± 0.08
R-medial rectus	0.92 ± 0.16	0.88 ± 0.17
R-lateral rectus	0.93 ± 0.04	0.88 ± 0.06
R-superior group	0.87 ± 0.14	0.80 ± 0.15
R-inferior rectus	0.93 ± 0.09	0.88 ± 0.11
All	0.92 ± 0.02	0.87 ± 0.03

**Table 4 diagnostics-12-01553-t004:** Performance of selected U-net on test data split by extraocular muscle regions (near insertion, muscle belly, and near origin).

Muscle	Region 1: Insertion		Region 2: Central Part		Region 3: Origin	
L-medial rectus	0.89 ± 0.13	0.82 ± 0.16	0.97 ± 0.01	0.94 ± 0.02	0.91 ± 0.10	0.85 ± 0.13
L-lateral rectus	0.88 ± 0.15	0.81 ± 0.15	0.95 ± 0.02	0.90 ± 0.03	0.94 ± 0.08	0.89 ± 0.09
L-superior group	0.79 ± 0.26	0.71 ± 0.25	0.92 ± 0.06	0.86 ± 0.07	0.92 ± 0.05	0.85 ± 0.08
L-inferior rectus	0.93 ± 0.04	0.88 ± 0.06	0.94 ± 0.07	0.89 ± 0.08	0.95 ± 0.02	0.90 ± 0.04
R-medial rectus	0.91 ± 0.16	0.85 ± 0.16	0.79 ± 0.34	0.75 ± 0.35	0.83 ± 0.25	0.77 ± 0.25
R-lateral rectus	0.77 ± 0.20	0.66 ± 0.21	0.93 ± 0.03	0.87 ± 0.05	0.94 ± 0.04	0.89 ± 0.06
R-superior group	0.78 ± 0.29	0.70 ± 0.28	0.91 ± 0.04	0.84 ± 0.06	0.89 ± 0.04	0.80 ± 0.07
R-inferior rectus	0.89 ± 0.16	0.83 ± 0.16	0.95 ± 0.07	0.90 ± 0.08	0.94 ± 0.03	0.88 ± 0.06
All	0.86 ± 0.20	0.78 ± 0.20	0.92 ± 0.14	0.87 ± 0.15	0.91 ± 0.11	0.86 ± 0.12

**Table 5 diagnostics-12-01553-t005:** Performance of selected U-net model (trained using WCE+Dice loss) on test data. Values indicate mean ± standard deviation of Dice score and IOU score on cross-validation data.

Muscle	MAE Thickness (mm)	MAPE Thickness	MAE Area (mm^2^)	MAPE Area
L-medial rectus	0.24	5%	1.99	6%
L-lateral rectus	0.35	7%	6.53	14%
L-superior group	0.37	8%	3.15	8%
L-inferior rectus	0.26	6%	4.2	10%
R-medial rectus	0.41	7%	3.93	8%
R-lateral rectus	0.46	9%	3.85	10%
R-superior group	0.33	7%	4.09	10%
R-inferior rectus	0.36	8%	3.18	9%
All	0.35	7%	3.87	9%

**Table 6 diagnostics-12-01553-t006:** Average performance of the trained U-net on 32 test cases where input images were added with different levels of Gaussian noise. Values indicate mean DSC (and standard deviation) and mean IOU (and standard deviation), respectively.

	Without Added Noise	With Added Noise (μ = 0, σ = 5)	With Added Noise (μ = 0, σ = 10)
L-medial rectus	0.94 ± 0.07	0.94 ± 0.08	0.93 ± 0.09
L-lateral rectus	0.93 ± 0.09	0.93 ± 0.07	0.92 ± 0.09
L-superior group	0.90 ± 0.12	0.89 ± 0.15	0.90 ± 0.13
L-inferior rectus	0.94 ± 0.08	0.94 ± 0.04	0.94 ± 0.09
R-medial rectus	0.92 ± 0.16	0.93 ± 0.10	0.92 ± 0.14
R-lateral rectus	0.93 ± 0.04	0.92 ± 0.08	0.92 ± 0.07
R-superior group	0.87 ± 0.14	0.86 ± 0.16	0.86 ± 0.17
R-inferior rectus	0.93 ± 0.09	0.93 ± 0.08	0.93 ± 0.07
All	0.92 ± 0.02	0.92 ± 0.12	0.92 ± 0.12

**Table 7 diagnostics-12-01553-t007:** Regional IOU score of our model and previously published CNN models for extraocular muscle segmentation on CT. Values indicate mean IOU *±* standard deviation. SU-Net and SV-net proposed by Zhu et al. [12] was trained and evaluated using images from 97 subjects without contrast enhancement and our model was trained and evaluated using images from 210 subjects with contrast enhancement.

Muscle	SU-Net	SV-Net	2D Coronal U-Net
Medial rectus	0.82 ± 2.83×10^-5^	0.84 ± 3.62 × 10^-5^	0.91 ± 0.12
Lateral rectus	0.80 ± 5.83 × 10^-5^	0.82 ± 3.56 × 10^-5^	0.89 ± 0.04
Superior rectus	0.73 ± 9.73 × 10^-5^	0.74 ± 7.84 × 10^-5^	-
Superior muscle group	-	-	0.84 ± 0.09
Inferior rectus	0.82 ± 2.83 × 10^-5^	0.84 ± 3.39 × 10^-5^	0.89 ± 0.06
Optic nerve	0.81 ± 1.77 × 10^-4^	0.82 ± 9.96 × 10^-5^	-
Total	0.80 ± 2.56 × 10^-5^	0.82 ± 3.22 × 10^-5^	0.88 ± 0.09

## Data Availability

The data presented in this study are available on request from the corresponding author.

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
