# Peer review of "Semantic Segmentation of Extraocular Muscles on Computed Tomography Images Using Convolutional Neural Networks"

_diagnostics, 2022, doi:10.3390/diagnostics12071553_

Round 1

Reviewer 1 Report

This paper applied CNN to analyze the CT images. The methodology is sound and the article is well written in general.

The reson of the application of U-net could be further discussed. There are a number of deep learning architectures out there, please give an improved review on similar CNN architectures in the existing works.

The results are good, however, the detailed settings of the CNN parameters can be further explained in order for the interested users to reproduce the results.

Author Response

We thank the reviewer for the constructive comments. We are pleased about the positive comments regarding the methodology and writing.  In the following, we describe the response to your comments.

  1. The reason of the application of U-net could be further discussed. There are a number of deep learning architectures out there, please give an improved review on similar CNN architectures in the existing works.

Thank you for the feedback. In the Materials and Methods section (section 2.1), lines 160 – 223, we’ve added review of CNN architectures used for medical image segmentation and we’ve also added details on why U-net has been used in this work

  1. The results are good; however, the detailed settings of the CNN parameters can be further explained in order for the interested users to reproduce the results.

In the Materials and Methods section (section 2.7), lines 373 – 400, we’ve added exact settings of CNN parameters that provided the best training & validation results. Also, in the Supplemental materials section, the codes have been made public and a hyperlink has been provided to facilitate interested users to reproduce the results

Reviewer 2 Report

The application studied in this paper is interesting and important. The paper is written well in most parts. However, there are some typos which needs correction. The paper looks like a short paper (or letter) because both the content and the results are very limitted. The contribution and novelty of the work is very limited. The following issues must be addressed before acceptance:

1. Introduction is too short. Motivation, contributions and novelty of the work are not clear.

2. Review of the related work is very limitted. There are several recent relevant works that have not been cited:

-https://assets.researchsquare.com/files/rs-1328096/v1/618b25eb-2605-4356-8ea9-c8d3493f7503.pdf?c=1645565665

-https://www.hindawi.com/journals/ije/2017/3196059/

-https://www.mdpi.com/948172

3. Figure 1 is incomplete.

4. Is the dataset public or private? Please provide a reference/access link if it is publicly available.

5. Typo in line 129: field

6. How robust your algorithm is against various noises? There is no mention of this. Also, no experiments on the performance of the proposed method against noisy images are given.

7. What do you mean by "your model" in line 168?

8. Any cross validation? please describe the details of the experiments in the results section.

9. How do the values of network parameters are selected. What are their effects on the perrformance?

10. The performance of this method must be compared and analysed against existing traditional segmentation methods.

11. The number of references is very small. That means the review of related works is not satisfactory.

Author Response

We thank the reviewer for the constructive comments. We are pleased about the positive comments regarding the application.   In the following, we describe the response to your comments.

  1. Introduction is too short. Motivation, contributions, and novelty of the work are not clear.

Thank you for the feedback. In the Introduction section (section 1), lines 46 – 111, we’ve added review of related work, challenges, novelty and our contributions.

  1. Review of the related work is very limited. There are several recent relevant works that have not been cited:

-https://assets.researchsquare.com/files/rs-1328096/v1/618b25eb-2605-4356-8ea9-c8d3493f7503.pdf?c=1645565665 

-https://www.hindawi.com/journals/ije/2017/3196059/

 -https://www.mdpi.com/948172

Thank you for the feedback. In the Introduction section (section 1), lines 46 – 111, we’ve added review of related work around the traditional algorithms and deep learning methods for extraocular muscle segmentation.

  1. Figure 1 is incomplete.

Figure 1 was intended to represent a typical convolutional layer of a CNN architecture. While the figure was complete, it appeared to be incomplete because of the way the illustration was formatted. We have edited the illustration to highlight the typical convolutional layer and used dotted lines to the format of the connection to the next layer.

  1. Is the dataset public or private? Please provide a reference/access link if it is publicly available.

Thank you for the question. The dataset is not public. This is mentioned in the “Data Availability Statement” section. The data presented in this study is available on request from the corresponding author.

  1. Typo in line 129: field

Thank you for pointing this out. The typo has been fixed to change ‘fField’ to ‘field’

  1. How robust your algorithm is against various noises? There is no mention of this. Also, no experiments on the performance of the proposed method against noisy images are given.

Thank you for your feedback. We updated the proposed U-net training process to include gaussian noise. Noise addition to CT images that can simulate low-dose acquisition settings require access to the raw scanning data [46, 47]. Since the raw sinogram data from the scanners was not available at the time of training, gaussian noise (of mean = 0 and standard deviation = 10 Hounsfield Units) was added to the CT slice intensities. In section 3.1.1, we present the performance of the trained model on images added with Gaussian noise.

  1. What do you mean by "your model" in line 168?

Thank you for this question. This line has been removed in the revised version we use cross-valiation instead of hold-out validation.

  1. Any cross validation? please describe the details of the experiments in the results section.

Thank you for this feedback. In the revised version, we have included cross-validation to improve reliability of model performance. In section 2.7, we have added the details of the experiments in lines 402-415, and the results in section 3.11 Table 2.

  1. How do the values of network parameters are selected. What are their effects on the performance?

We present the cross-validation results that compare the performance of the model trained with various network parameters including the loss function. The effects on the performance are presented in section 3.11, Table 2. The rationale behind choosing the parameters is elaborated in section 2.7 from lines 374-400.

  1. The performance of this method must be compared and analyzed against existing traditional segmentation methods. Add evaluation criteria to match what they had

We added Intersection over Union as an evaluation criterion to present an analysis of existing segmentation methods. However, the imaging studies and ground truths that were used varied between the studies. For this reason, we

  1. The number of references is very small. That means the review of related works is not satisfactory.

Thank you for your feedback. We have added the references of the related works along the lines of automated segmentation using deep learning, measurement methodologies for thickness and area.

Round 2

Reviewer 2 Report

I am happy with the revision made by the authors.

Author Response

Thanks